# Museum Tourism 2.0: Experiences and Satisfaction with Shopping at the National Gallery in London

**Jun Shao [1], Qinlin Ying [2], Shujin Shu [1], Alastair M. Morrison [3,\*] and Elizabeth Booth [3]**

[1]  School of Landscape Architecture, Beijing Forestry University, No.35 Tsinghua East Road, Beijing 100083, China; ninashaojun@bjfu.edu.cn (J.S.); sueshushujin@bjfu.edu.cn (S.S.)

[2]  Dornsife College of Letters and Arts and Science, University of Southern California, 3551 University Ave, Los Angeles, CA 90089, USA; qinlinyi@usc.edu

[3]  Business School, Department of Marketing, Events and Tourism, University of Greenwich, London SE10 9LS, UK; e.booth@gre.ac.uk

\*  Correspondence: a.morrison@gre.ac.uk

**Abstract:** The tourist shopping experience is the sum of the satisfaction or dissatisfaction from the individual attributes of purchased products and services. With the popularity of the Internet and travel review websites, more people choose to upload their tour experiences on their favorite social media platforms, which can influence another's travel planning and choices. However, there have been few investigations of social media reviews of tourist shopping experiences and especially of satisfaction with museum tourism shopping. This research analyzed the user-generated reviews of the National Gallery (NG) in London written in the English language on TripAdvisor to learn more about tourist shopping experience in museums. The Latent Dirichlet Allocation (LDA) topic model was used to discover the underlying themes of online reviews and keywords related to these shopping experiences. Sentiment analysis based on a purpose-developed dictionary was conducted to explore the dissatisfying aspects of tourist shopping experiences. The results provide a framework for museums to improve shopping experiences and enhance their future development.

**Keywords:** The National Gallery; museum tourism; shopping experiences; tourist satisfaction; TripAdvisor; social media; UK

## 1. Introduction

Museum tourism is an important showcase for displaying cities' unique cultures and histories [1]. Tourist shopping experiences reflect the satisfaction or dissatisfaction gained from the attributes of purchased products and services [2]. Museums are transforming from cabinets of curiosities, meaning the custodians of collections, to cultural shopping experiences as a tool of economic transformation and a part of the tourism infrastructure [3]. Current debates revolve around the idea of the museum as a 'cultural shop' [3–6], a place where visitors come to enjoy, participate in, or consume a variety of educational and cultural products and merchandise. Although there are many studies on shopping experiences and satisfaction in tourism [2,7,8], the shopping element of museum tourism has not yet received much attention.

The increasingly pervasive culture of consumption and growing economic constraints have made the roles of museum tourism more complex and demand-oriented. Museum visitors are experiencing this evolution of museums' role; increasingly, visitors have transformed from 'spectators' to 'cultural shoppers' during trips to museums [3]. The increasing orientation towards income generation by museums is a trend of growing attention to social, recreational, and participatory experiences, redirecting the traditional and singular focus on collections and exhibitions [9]. In terms of participatory

cultural experiences, visitors can attend learning programs organized by museums and interact with instructors [9]. Sociability refers to the experience that both visitors and members seek. Museums hold events to meet social needs. For example, museums can have more seating, social spaces for members, dining facilities, and even grandparents' rooms. In addition, the recreational experience is an important element of the museum [5,9].

Moreover, with the rapid growth of digitalization and informatization as well as the propagation of consumer commentaries, many museums visitors write reviews and post them on social media. Consumers create authentic content and share their positive and/or negative emotional experiences on social media [10]. These trends make it possible to collect large amounts of data from various social networks to evaluate corporate performance, improve customer experience, and identify opportunities for service innovation from the customer's perspective [10–12].

This research analyzed online reviews from TripAdvisor (www.tripadvisor.com), the largest social travel website in the world, with about 315 million reviewers and over 730 million reviews of hotels, restaurants, attractions, airlines, and other travel-related businesses [13] under the four-dimension experience framework proposed by Kotler [9].

The following research questions were addressed:

1.  What are the underlying themes of online reviews about museums related to shopping experiences?
2.  What are the dissatisfying aspects of tourists' shopping experiences during their museum tours?

The National Gallery in London is one of the UK's flagship visitor museums and was used as the case study for this research. It is highly commercialized and contributes significantly to the tourism economy of the city.

## 2. Literature Review

### 2.1. Museum Tourism

With the development of urban destinations, the tourism function of museums has become increasingly prominent. Museum tourism is also receiving more attention from scholars. At present, museum tourism research focuses on the evolution of the roles and functions of museums, the behavior of museum visitors, and the development of museum tourism.

In terms of museum functions, most scholars agree that museums have developed into multi-functional institutions integrating education, leisure, entertainment, and social significance rather than merely providing the collection, protection, and exhibition of cultural and spiritual heritage [9,14–17]. Kotler explained that, in addition to providing educational and display functions, museums now present a trend of providing participatory, social, and recreational experiences. Museums are becoming multi-functional places where recreational and learning experiences are combined, allowing visitors to stroll around galleries and view exhibitions under an intense entertainment sensory stimulation [9] (Kotler, 2004).

Experts also believe that the integration of museums with cities and communities have become a trend [18,19]. Researchers are increasingly exploring the motivations of museum visitors as well as their needs, experiences and satisfaction, as well as their behavioral characteristics. For example, McLean found that most people's motivation in visiting a museum was to gain more cultural knowledge to enrich their life experiences [20]. Both in studying the characteristics and composition of visitors to the Science Museum in London, concluded that various groups had different needs with respect to museums [21]. The current research on museum tourism development focuses on the design and creation of resources and products as well as the current status and future trends in museum tourism. For example, Chen, Lee, Lin, and Wang used the decision-making trial and evaluation laboratory method to explore the critical factors influencing visitors to purchase museum cultural products [22].

Generally, although current research on museum tourism is expanding, there are relatively few studies on the shopping experiences and satisfaction of museum visitors. The purpose of this research is to enrich museum tourism literature by investigating the shopping experience of museum visitors.

## 2.2. Shopping Experiences in Museums

Shopping is one of the important tourist activities and is acknowledged as a primary travel motive [8,23,24]. Given its importance, more researchers are paying attention to the shopping experience within tourism. However, museum shopping experiences have received scant attention. Museums are an integral component of cultural heritage and are important heritage attractions, especially within major cities [25]. If visiting museums is regarded as a form of tourism, the shopping experiences of museum visitors may bear some similarities to other tourism shopping experiences [26]. Below is a review of the related literature on general tourist shopping experiences.

The above-mentioned definition of shopping tourism developed by Tosun et al. [2] has been confirmed by many scholars who have found that satisfaction with a series of shopping attributes such as products, services, and shopping environment leads to overall satisfaction with shopping experiences [2,7,8]. For example, Wong and Wan defined tourist shopping satisfaction as a four-dimensional construct, including service product and environment, merchandise value, staff service quality, and service differentiation. They pointed out that positive assessments of these dimensions generate pleasant shopping experiences for visitors [8]. Heung and Cheng divided shopping attributes into four dimensions: staff service quality, product value, product reliability, and tangible quality. They found that staff service quality had the greatest effect on visitor satisfaction when shopping in Hong Kong, followed by product value and product reliability [7].

Similarly, museum researchers recognize that many factors affect the quality of and satisfaction with visitor experiences. Falk and Dierking proposed an interactive experience model, believing that visitor experiences with museums are generated by the interaction of the physical environment, personal background, and social environment [27]. Rowley believed that visitor experiences in museums are pervasive and identified 10 factors that influence such experiences, namely speed of service delivery, convenience, age waves, choice, lifestyle, discounting, value adding, customer service, technology, and quality [28]. For city museums, Goulding (2000) thought that the service experience was influenced by a number of socio-cultural, cognitive, psychological orientations, and physical and environmental conditions [29]. Forrest found atmospherics to be a significant aspect of visitor experiences in museums. The museum environment also affects visitor satisfaction [30]. Jeong and Lee found that, among three factors of the museum physical environment, the exhibition environment had the greatest effect on visitor satisfaction. The size of the museum had a slight direct effect on satisfaction, while the ambient environment had an indirect effect [31].

The role of museum retail spaces has been redefined, becoming a more important element of museum visits [3]. Kent argued that museum shops play an important role in creating interactivity and recreational experiences that supplement the museum's educational priorities [32]. But he did not consider museum shops as venues that satisfy the sociability needs of members as Kotler [9] discussed. This research draws upon these points of museum retailing to further analyze visitors' cultural shopping experiences and satisfaction.

## 2.3. Content Analysis in Museum Tourism

Content analysis has increasingly been applied to tourism research [33] and recently with user-generated content (UGC) on social media [34].

Content analysis is a research method that converts otherwise qualitative and symbolic contents, such as photographs and text, into systematic and quantitative data [35]. It is widely used to analyze behavioral patterns, tourism experiences, and destination image perceptions of visitors drawn from online content. For example, Choi, Lehto, and Morrison identified the image representative of Macao on the Internet by analyzing the contents of various online information resources, including Macao's official tourism website, tourism blogs, and travel agency websites [36]. Banyai found that Macao's image in tourism as described by Western tourists in blogs was inconsistent with that created by local tour guides, and he proposed improvements in the tourism market strategy for Macao according to these differences [37].

Some scholars have begun to use content analysis to analyze museum reviews online travel websites, especially those on TripAdvisor [38]. For example, Carter analyzed 200 TripAdvisor reviews on the Southern Plantation Museum of the United States [39]. Souto analyzed 1007 tourist reviews about the Berlin Museum on TripAdvisor [40]. Su and Teng (2018) studied museum service failures based on 301 negative comments from 15 different national museums [41]. These studies demonstrate the effectiveness of online platforms as data sources and the application of content analysis to analyze UGC. This investigation used content analysis of TripAdvisor's reviews to analyze visitor shopping experiences and satisfaction while at the National Gallery.

There are many types of sentiment analysis techniques that have been widely used in the social media domain, such as the Linguistic Inquiry and Word Count (LIWC) [42] and the Valence Aware Dictionary and Sentiment Reasoner (VADER) [43]. As a parsimonious rule-based model for sentiment analysis of social media text, VADER's effectiveness outperforms and generalizes more favorably across contexts than other state-of-practice benchmarks including LIWC, ANEW, the General Inquirer, SentiWordNet, and machine learning oriented techniques relying on Naive Bayes, Maximum Entropy, and Support Vector Machine (SVM) algorithms [43]. Therefore, this research chooses VADER to investigate the dissatisfying aspects of online reviews.

## 3. Methodology

To measure visitors' experiences and their satisfaction with a museum, this study used data mining, text analysis, and sentiment analysis techniques provided by the Python Natural Language Toolkit (NLTK) supplemented to thick data analysis. Online reviews of visitors to the National Gallery (NG) were collected from TripAdvisor.com. Figure 1 shows the data analysis process for this study.

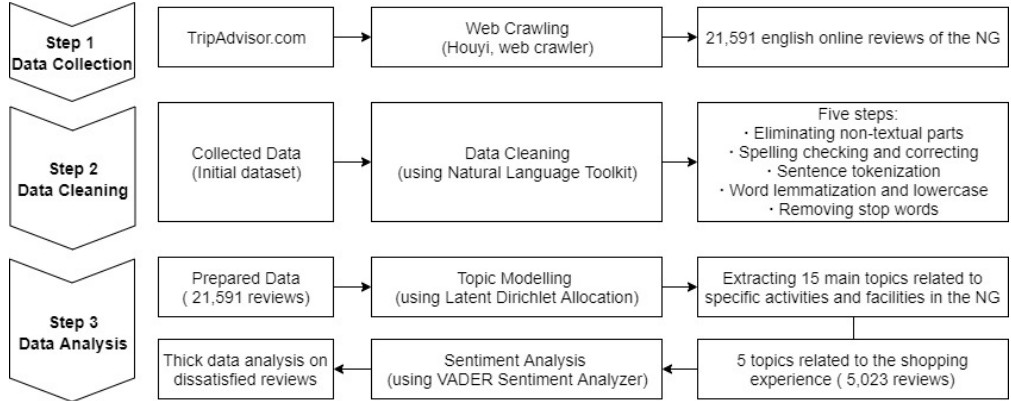

**Figure 1.** Process of data analysis.

### 3.1. The National Gallery

The National Gallery (NG) is an art museum located at Trafalgar Square in Central London. Founded in 1824, it houses a collection of over 2300 paintings dating from the mid-thirteenth century to the 1900s [44]. It is one of the UK's most representative nationally-funded museums; it is also a non-departmental public body (NDPB), whose sponsoring body is the Department for Culture, Media and Sport [45]. It is regarded as a 'super museum' and is widely known and popular around the world as it houses a large collection of internationally renowned personal paintings and works by famous artists. The NG has been commercialized on a large scale, making a significant contribution to London's tourism economy [46]. As the culture of consumption has increasingly become ingrained in Western society and economic constraints have led to cuts in government funding for culture and heritage, the NG is now adept at generating income from trade and other sources [3].

According to the NG's official website (https://www.nationalgallery.org.uk/), the shopping experiences it offers include the process of purchasing products and services as follows. In terms of

merchandise for sale, the NG has three shops that sell high-quality gifts including books, postcards, stationery, t-shirts, and jewelry. The stores have a wide selection of art books and gifts related to the gallery's incredible collections, which cannot be found elsewhere. The NG also has restaurants and coffee shops for visitors. The museum does not require visitors to pay for admission; tickets are not required although donations are accepted. For visitors who worry about getting lost in such a huge gallery, they can buy gallery maps and enjoy wonderful journeys. Those desiring a deeper understanding of the stories behind the masterpieces can rent audio guides. The audio guides provide an interpretation of an more than 80 paintings in various languages. The NG regularly holds special paid exhibitions such as "The Retrospective of Monet and Architecture", which showcased more than 70 architectural paintings by Monet for the first time. Membership is available at £68 for an individual member and £107 for one individual member and one guest. NG members enjoy benefits such as free admission to exhibitions, preview days, members-only viewing hours, priority booking for public events, and discounts at NG stores.

The overview of online reviews related to the NG on TripAdvisor is shown in Figure 2.

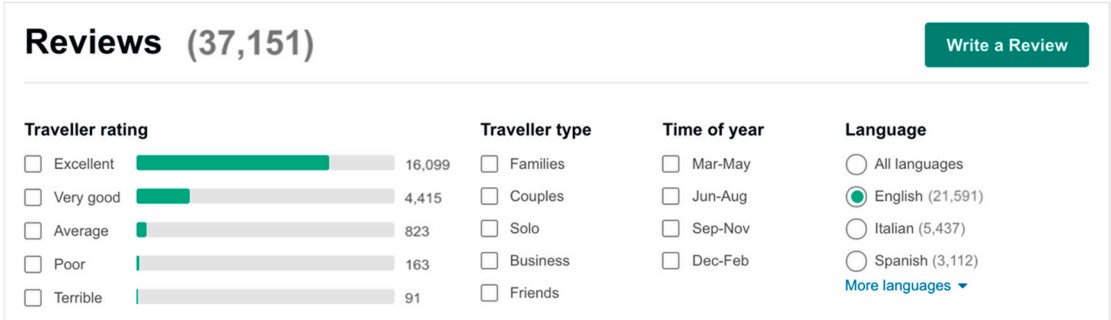

**Figure 2.** Overview of visitor reviews on the National Gallery on TripAdvisor.

*3.2. Data Collection and Preprocess*

As shown in Figure 1, the Houyi, a web crawler tool, was applied to capture online comments about the NG in English from TripAdvisor on 13 March 2019. First, via a search of the homepage of TripAdvisor using "The National Gallery" as the keywords, there were 37,151 reviews related to NG from 5 February 2004 to 13 March 2019. Next, 21,591 reviews written in English, accounting for 58.2% of the total, were selected by choosing the English option box of the search engine offered by TripAdvisor. However, some comments with more than 45 words were only partially shown on the search results page and a circular crawl list was created by the researchers to get the full details in each review. Then, the full contents of each review were extracted according to the page structure. Finally, all relevant online reviews by NG visitors (over 700,000 words in English) were collected from TripAdvisor by the automatic page-turning function of the software.

In the data/text preprocessing step, this research used Python NLTK to eliminate non-textual parts, check spelling, introduce sentence tokenization, word lemmatization, and remove "stop words" like "the", "a", "on", "is", and "all". After these steps, 21,591 reviews remained for further analysis.

## 4. Data Analysis and Results

For text analysis, the LDA topic modeling algorithm was used to mine themes of reviews. Then, reviews belonging to the topics related to shopping experiences were further classified according to the four dimensions proposed by Kotler [9]. The VADER Sentiment Analyzer designed by Hutto and Gilbert [43] in Python NLTK was used to analyze emotions in the online reviews. Furthermore, a process of thick data analysis was conducted on negative comments to find out the dissatisfying aspects of the NG.

## 4.1. Topic Modelling Analysis of Online Reviews

Topic modeling is a powerful algorithm for discovering hidden structures in large text sets. It is widely used in natural language processing, text mining, social media analysis, information retrieval, and other fields [47]. LDA was proposed by Pritchard, Stephens, and Donnelly [48] and is a widely applied approach in topic modeling. In natural language processing, LDA is a generative probabilistic model of the corpus: it represents a document as a random mixture of latent topics, in which each topic is characterized by a distribution over words [49]. LDA was used to extract the topics most expressed by NG visitors and the probability of these topics from online reviews. The LDA modelling on the NG data revealed the 15 most important topics and the top 10 words for each topic. Topic naming relies on identifying logical connections between the most common words in the topic [50]. This is a moderately abstract way of naming topics based on top-level keywords. For example, in Table 1, the topic of 'amazing collections' is based on the words 'amazing' (22.5%), 'Van Gogh' (12.8%), and 'Monet' (9.8%), which made up the largest portion of the topic.

**Table 1.** Examples of topic naming.

| Topic | % | Topic | % |
|---|---|---|---|
| **Topic 1: Amazing Collections** | | **Topic 2: Building** | |
| Amazing | 22.5% | Gallery | 81.2% |
| Van Gogh | 12.8% | Lover | 4.0% |
| Monet | 9.8% | Person | 2.9% |
| Audio | 9.7% | Lost | 1.4% |
| Masterpiece | 7.6% | History | 1.3% |
| Experience | 7.5% | Color | 0.9% |
| Include | 3.7% | Massive | 0.9% |
| Gallery | 3.6% | Lucky | 0.9% |
| Classical | 3.2% | Focus | 0.7% |
| Decide | 2.9% | Fortune | 0.4% |

Table 2 shows the 15 most significant topics mined from the 21,591 online reviews. The top nine themes with a high proportion of topics were related to specific activities and facilities in the NG, including exhibitions, admission, free programs, building, paid programs, stuff, shops, tickets, and dining facilities. Two of the topics showed overall how visitors rated the NG, 'amazing collections' and 'crowded tours'. On the one hand, the 'amazing collections' reflect surprise and amazement with the massive collections. On the other hand, the 'crowded tour' implies that some visitors experienced crowding in the NG. Other topics represent essential information on the NG that visitors are concerned about, including 'time to visit', 'length of visit', 'location', and 'layout'.

**Table 2.** Distribution of topics.

| Topic | Proportion | Topic | Proportion |
|---|---|---|---|
| Exhibitions in gallery | 7.2% | Dining facilities | 6.4% |
| Admission | 7.2% | Amazing collections | 6.3% |
| Free programs | 7.2% | Length of visit | 6.3% |
| Building | 7.2% | Location | 6.2% |
| Paid programs | 7.2% | Layout | 6.0% |
| Stuff | 7.1% | Time to visit | 6.0% |
| Shop | 7.1% | Crowded tour | 5.8% |
| Tickets for exhibitions | 6.9% | | |

Overall, these 15 topics provide a good overview of the NG's online reviews on TripAdvisor. The demands of tourists are very important and these are expressed in some of their open-ended comments, such as catering facilities, layout, architecture, programs, exhibitions, tickets, and shops.

The presence of negative evaluations should not be ignored, such as 'crowded tours', as the overall experiences are more significant than the exhibits these visitors.

### 4.2. Visitor Shopping Experiences in Online Reviews

As mentioned above, Kotler [9] suggests that museums provide social, recreational, and participatory experiences beyond the educational and intellectual. Accordingly, four dimensions of museums' tourism experience can be defined as educational, social, recreational, and participatory. The tourist shopping experience provided by the NG's website (URL: https://www.nationalgallery.org.uk/) was classified according to these four dimensions, as seen in Table 3. It is worth noting that some activities include multiple experiences. For example, paid exhibitions provide some educational experiences for visitors, while some of the activities included in a paid exhibition can provide visitors with more recreational experiences.

**Table 3.** Four dimensions of museum tourism experiences.

| Dimensions | Definition | Related Activities in Official Website |
|---|---|---|
| Educational | The activities that enable people to gain knowledge | Paid audio guides; paid exhibitions |
| Social | The activities which satisfy people's social needs | Catering facilities; souvenir shop; memberships |
| Recreational | The activities which enable people to gain enjoyment and happiness | Paid exhibitions |
| Participatory | The activities in which enable people immerse themselves | Memberships; paid exhibitions |

Among the 15 topics identified by LDA, five topics were directly related to the shopping experiences of visitors, namely 'dining facilities' (keywords: restaurant, location, display, dining, ticket, British, minute, tea, late, tire); 'paid programs' (keywords: cost, spend, build, impress, treasure, disappoint, culture, pay, short, opportunity); 'stuff' (keywords: national, staff, shop, special, overwhelming, outstanding, original, hand, story); 'shop' (keywords: collection, master, shop, offer, set, quality, choose, citizen, bring, Western); and 'tickets for exhibitions' (keywords: time, worth, ticket, view, visitor, charge, pick, enter, leave). In total, 5023 original comments contained these shopping experience keywords. Thick data analysis was applied to analyze these reviews, searching each keyword and taking the most relevant 50 results. The results show that many shopping experience activities were not mentioned on the official website, such as paid books, paid maps, games in shops, and interactive activities (Table 4).

**Table 4.** Activities in the National Gallery (NG).

| Dimension | Activities in NG's Official Website | Activities Only in Online Reviews |
|---|---|---|
| Educational | Paid audio guides; paid exhibitions | Paid books; paid maps |
| Social | Catering facilities; souvenir shop; memberships | None |
| Recreational | Paid exhibitions | Games in shops |
| Participatory | Memberships; paid exhibitions | Interactive activities |

Next, these keywords and their original reviews were further classified according to the four dimensions proposed by Kotler [9]. The classification results showed that the visitors' shopping experiences in the NG mainly related to the guide/audio guides, exhibitions, books, maps, gift shop, games, memberships, and catering services (Table 5).

**Table 5.** Specific contents on shopping experiences in the NG.

| Dimension | Keywords | Online Reviews on TripAdvisor |
| --- | --- | --- |
| Educational | Guide, audio-guide | Invest in an audio-guide to fully explore the museum. |
| | | Great exhibition started on the 18 of March I suggest to take Audio-guide to get more information. Cost £16 Adult under £12 goes for free if you pay online you £2 discount and avoid weekends as it is more expensive. |
| | | The audio-guide is definitely worth it, the stories behind some of the paintings were really interesting. |
| | Exhibitions | Extensive art collection (free entry) and special exhibitions (extra fee) welcome you to indulge in cultural experience of the extraordinary kind. |
| | | Fantastic Gallery, the paid exhibitions are also worth seeing! |
| | | The joys of the permanent collection are well documented, so I will focus on the recent "paid for" exhibition: Sorolla: Master of Light. Simply outstanding. |
| | Book | The 144 pages book with all exhibited works and photos from Courtaulds Home House is also excellent. |
| | | If you take children do buy the Katie books as it really brings the pictures to life for them. |
| | | There are so many beautiful paintings to see, so you really need to buy one of the gallery books. |
| | Map | We stopped in at the National Gallery to pick up a map (maps cost 1 pound) for a future visit which was really worth. |
| | | For two pounds you can buy a museum map that highlights some of the more famous pieces to see if you have only an hour. |
| Social | Lunch, cafeteria, restaurant | The cafeteria area and loos are well managed and not too expensive for a treat an unmissable landmark. The restaurant is a fabulous spot for lunch with exceptional views of the city and a small menu very inviting and yummy. |
| | | The restaurant is a fabulous spot for lunch with exceptional views of the city and a small menu very inviting and yummy. |
| | Tea, cake, coffee | Cafe is good, we sat in the bar and had a great coffee and tea in very atmospheric surroundings. |
| | | The excellent interactive planning stations in the coffee shop/basement allow you to easily find any particular painting or artist. |
| | Gift shop | Large gift shop with good selection of books and prints for sale. |
| | | The shop is a great place for finding interesting gifts. |
| Recreational | Games | There are a lot of entertainment for kinds. There is shop with souvenirs and games. |
| | | Don't miss Holbein's The Ambassadors and enjoy the perspective skull game! |
| Participatory | Member-ships | I have been a member for a year and have enjoyed the free entry to exhibitions and the reduction in the cost of learning courses that this offers. The membership is a good way of contributing to and reaping massive rewards from this wonderful art collection. |
| | | The great thing about the National Gallery being free to visit (we pay nearly £100 a year for membership, so don't feel the need to make additional voluntary donations) is that you can slip in for a few minutes, gaze at just one astounding work and leave feeling a little better about life. |

*4.3. Satisfaction with Shopping Experiences in the National Gallery (NG)*

The VADER Sentiment Analyzer analyzes the emotions of each comment, scores the emotions ranging from −1 to 1, and classifies the positive, negative, and neutral polarity of the emotions according to the scores. If the compound score is less than 0, the review is negative; however, if the

value of compound is more than 0, the review is positive [43]. Table 6 shows the results of the VADER sentiment analysis applied to the entire set of online reviews related to shopping experiences: there were 4689 positive (93.4%), 110 neutral (2.2%), and 224 negative reviews (4.5%). This indicated that visitor sentiment towards the NG was mainly positive. To better understand the sentiment intensity of reviews, the absolute values of compound scores were divided into five levels; the value in [0.8,1] was set as 'very intense', [0.6,0.8] as 'intense', [0.4,0.6] as 'medium', [0.2,0.4] as 'weak', and [0,0.2] as 'very weak'. The results showed that the 'very intense' positive reviews accounted for 60.2% of total positive reviews and the 'very intense' negative reviews accounted for 4.5% of total negative reviews.

**Table 6.** Emotional analysis of visitors' online comments of the NG.

| Category | Total Number | High Intense | Intense | Medium | Weak | High Weak |
|---|---|---|---|---|---|---|
| | | Number (Proportion) | Number (Proportion) | Number (Proportion) | Number (Proportion) | Number (Proportion) |
| Positive | 4689 | 2824 (60.2%) | 1023 (21.8%) | 511 (10.9%) | 226 (4.8%) | 105 (2.2%) |
| Neutral | 110 | 0 | 0 | 0 | 0 | 0 |
| Negative | 224 | 10 (4.5%) | 32 (14.3%) | 35 (15.6%) | 68 (30.4%) | 79 (35.3%) |

The process of thick data analysis on dissatisfied comments was as follows: select the top 100 online comments with the lowest scores for negative emotions and classify them according to the four dimensions of "education, social, entertainment, participation" proposed by Kolter [9]. As shown in Table 7, dissatisfaction with the NG focused on poor and sub-standard guide service, overpriced paid exhibitions, inconsistent content and prices, overcrowding, poor catering services, food with low cost performance, untidy public facilities, too much noise, lack of interactive activities, and inadequate membership system.

**Table 7.** Dissatisfaction evaluation of NG visitors.

| Dimensions | Keywords | Existing Problems | Online Reviews on TripAdvisor |
|---|---|---|---|
| Educational | Staff | Poor service | First of all, many of the room guides were talking to one another, which makes them less approachable as you have to essentially break up their conversation in order to ask them a question . . . . So the room guides are not trained to know about the works/artists on display? They seem only to be able to tell you what room an artwork is in . . . .<br>When I said this was a poor show, she told me if we were that bothered to see the exhibition it had been open since 10 am, gave us a final sneer and turned away. This is a major national institution that is treating its supporters like disposable trash! In this day and age, I believed rubbish customer service was a thing of the past. Pull your socks up NG or I'll be relinquishing my membership. I would have sent this direct to NG leadership but can't see a place for feedback on the website. |
| | | Rudeness | Staff are rude and abrupt and really aren't customer friendly at all. All in all I pleasant and interesting place but the staff need some customer training. |

**Table 7.** *Cont.*

| Dimensions | Keywords | Existing Problems | Online Reviews on TripAdvisor |
|---|---|---|---|
| | Paid exhibition | Charge too high and beyond people's expectations | The exhibition is not worth the 18 GBP asking price. The majority of the works are by Sebastiano—A magnificent artist. But Michelangelo gets top billing here, and where there are Michelangelo works, many are drawings and studies. What's worse is that the sculptures are replicas. I understand that there is no way to bring the Pieta from Italy to London, but to have a replica in its stead, and charge such a high price of admission is a sham. I found this one incredibly poor of contents, the price ridiculously high (£16 adults) and who create the panels and the labels?<br>It was very disappointing and not worth the money £14. It's only 6 rooms and all of them include only Caravaggio inspired artists. Only few painting by him would be available for free in the National Gallery itself or in National Gallery in Dublin. |
| | | Disappointing content | I usually very much enjoy National Gallery exhibitions—We are members and visit often. However the current Caravaggio exhibition is a disgrace in my view. You cannot market such an exhibition with only 6 pictures by the named artist!! It's not that the others are not good, they are, especially the Ribera's but it's really shocking that so much is made of the influence of Caravaggio with so few examples.<br>Do you know its quite annoying to read bright white words on a dark base...?! I felt dizzy through all the exhibition. |
| | | Too many tickets for sale | Great exhibition but ruined by overcrowding caused by the National Gallery allowing too many visitors to buy tickets given the space and time available. However, I was a little overwhelmed by the number of people crammed into the basement levels, considering that the tickets were on sale as timed entry, which would make you believe that safe numbers of people had been calculated. I doubt that very much. |
| | Map | Paid but not very useful | The gallery is very confusing. Even the gallery map is very useless. I just say for almost 30 min and I left. Total waste of time!! |

**Table 7.** *Cont.*

| Dimensions | Keywords | Existing Problems | Online Reviews on TripAdvisor |
|---|---|---|---|
| Social | Dining facility | Poor service | Staff were rushing about looking hassled and the head waiter was joking and chatting to another waiter but not serving the tables. I finished my coffee and then waited a further 30 min before someone was available to bring my bill. I left feeling frustrated at the quality of service and will not go back. The service was terrible—I had to go and find a waiter because they seemed to have forgotten us. I'd tried to telephone ahead to see if we needed to make a booking but couldn't get through until finally someone, who spoke very little English, said "only reservation for restaurant" and put the phone down. |
| | | Expensive and not delicious | The cakes are supposedly freshly made but the Bakewell tart was stale and dry and the other cakes were just plain dull. Awful. Thought we may get ordinary fish and chips but it was disgusting, overlooked and tasted strange. Partner could not eat hers. |
| | Gift shop | Poor service | I was on my phone trying to confirm which postcards my friend wanted when the staff took the postcards and an art print directly away from my hands and told me off the shop. |
| | Service facilities | Unclear | Visited the National Gallery on 26 July 2017 the ladies toilets were appalling. Wet floor, out of order sinks and general untidiness. Not good for a building with a large tourist capacity. Also, tea room in basement left a lot to be desired with uncleared and dirty tables. It is just as important to keep the public facilities clean and inviting especially when after viewing exhibitions you want to freshen up and relax. |
| Recreational | Atmosphere | Noisy | I am giving up on trying this gallery. The art is wonderful but the atmosphere is not one to appreciate any of the masterpieces. Lots of selfies and endless photo snapping, spoil quiet contemplation and meaningful study of some of the finest art in the world. A real shame, it is deeply depressing but no one cares. |

**Table 7.** *Cont.*

| Dimensions | Keywords | Existing Problems | Online Reviews on TripAdvisor |
|---|---|---|---|
| Participatory | Member-ships | Tickets to members only | Just went to book a ticket for the Monet exhibition, only to find that they are now only allocating tickets to members. So a £20 ticket suddenly becomes a £54 ticket. Greedy, greedy, greedy. Price gouging just because it's the last few days of the exhibition! Shocking behaviour. |
| | | Influencing non-members' visiting experience | The National Gallery has assembled a superb exhibition of Goya portraits. However when we visited, supposedly on a 'members' evening' it was almost impossible to breathe, let alone move independently in the exhibition—They just allowed too many people in. |
| | | Not worth it | The National Gallery is one of the UK's most prized treasures but do not make the mistake of joining as a member. We did and travelled down from Newcastle for the Rembrandt exhibition only to find that no priority is given to members. At 9.45 a.m. we had to join one queue of over 300 people trying to gain entry for the 10 a.m. opening and once inside the members queue was so long, they estimated it would take another hour and a half to gain entry . . . We visited the Uffizi and Prado last month and had no such issues. |
| | Interactive activity | little and boring | Not a place for children. We went on a school tour, thinking they may do something interactive but all they got was an hour of talking to about paintings! Even the adults were bored! The children asked when they were leaving & was this it, & how bored they were.<br>Went with kids 3 and 9. No interactive exhibits. The usher said there was a kids kit that can be picked up but it required a long walk inside the gallery to retrieve. Had to leave after 10 min because the kids were getting antsy. |

## 5. Discussion and Conclusions

### 5.1. Conclusions

This study investigated the underlying themes of online reviews on museums related to the shopping experiences; examined the dissatisfying aspects of tourists' shopping experiences during their museum tours; and applied big data and natural language analysis techniques as an innovative method. To achieve these goals, content analysis of visitor reviews on museums in social media was used. The results indicated that there are five most significant topics directly related to the shopping experiences, namely 'dining facilities', 'paid programs', 'stuff', 'shop', and 'tickets for exhibitions.' Visitors care about the educational (guide/audio guides, exhibitions, books), social (gift shops and catering services); recreational (games), and participatory (memberships) dimensions of their experiences. Furthermore, given that visitor sentiment towards the NG was mainly positive,

museum visitors' complaints about their shopping experiences mainly focused on the cost-value of the paid-for products and services, staff service attitudes, and the environmental facilities of the museum.

## 5.2. Theoretical Implications

This research is among the first to employ content analysis based on social media to examine visitor shopping experience in museums, particularly in its use of LDA modelling and VADER sentiment analysis. Specifically, the LDA topic modelling analysis of online reviews presents an overview of visitor concerns and some shopping-related topics that are not shown on the official NG website. The VADER sentiment analysis revealed the emotional tendencies of visitors as well as their satisfaction with shopping experiences in the NG.

Because visitor shopping experiences play an important role in contemporary museum tourism, the findings will reinforce previous shopping tourism studies for the benefit of museum tourism research. The increasing demand for cultural shopping in museums underlines the need for finding more effective ways to understand visitors' specific needs and thereby enhance their satisfaction levels. Museums should apply the recommended approach derived from the results to serve their attendees better. For museum researchers, this analysis helps uncover critical areas in shopping and service offers directly from visitor feedback that heretofore were not adequately exposed. Therefore, a new conceptualization of museum shopping experiences derived from the museum experiences model proposed by Kotler [9] can be developed.

## 5.3. Practical Implications

This research is of certain significance for the future development of the museum and the improvement of visitors' shopping experiences. The results show that visitors' shopping experiences in NG are related to many factors. Among them, the factors that worsen shopping experiences mainly include the cost-value of the paid products and services, staff service attitudes, and the environmental facilities of the museum.

In terms of the paid products and services, visitors are very concerned about the price–value–performance ratios of the products and services they purchase, which directly affects their shopping experiences. Furthermore, this research found that when visitors enjoyed paid products and services, they expected to get enhanced experiences related to their purchases. The revenue from paid products and services accounts for a sizeable part of the museum's income; similarly, products and services valued by tourists can expand the viability and influence of the museum. Therefore, museums should pay attention to and constantly optimize the design of paid-for products and services to enhance visitor experiences and satisfaction.

The findings of this research suggest that the staff service attitudes are important in affecting the shopping experiences of visitors. The carelessness of the tour guides, lack of respect for visitors, and inability to answer questions from visitors not only reduced the quality of shopping experiences but also left visitors with a bad impression of the NG. Therefore, the museum should improve the quality of its staff, which can be achieved by strengthening training and establishing a service culture and feedback mechanism.

Third, the research shows that the environmental facilities in the museum also have a significant impact on the shopping experiences of visitors. This is reflected in the cleanliness of facilities, a quiet atmosphere of the museum, and an absence of crowding. Accordingly, museums should ensure the comfort and cleanliness of facilities at all times. During peak periods, museums need to take some specific measures, such as limiting the number of visitors every day and offering different priced tickets, to ensure that visitors get an optimum experience.

## 5.4. Limitations and Future Research

Although this research illustrates the shopping experiences and satisfaction of tourists in the NG through social media data, it still has certain limitations, which can provide directions for future

research. First, this study only selected TripAdvisor as the data source to collect online reviews, which may have a platform bias. Therefore, it is recommended that this method be used for comments from other available sources, such as Facebook, Twitter, and Instagram.

Second, this investigation did not analyze the demographic characteristics of the online reviewers nor did it further discuss the demographic differences in the shopping experiences of tourists. Therefore, studying the shopping experiences of different visitors in the future is an area worth exploring.

Third, although LDA is a good topic modeling algorithm for extracting topics from online comments, it still has some limitations. For example, LDA does not consider the interposition of words in the document. Documents like "Man, I love this can" and "I can love this Man" may be modeled in the same way [51]. At the same time, LDA cannot accommodate for the phenomenon of polysemy of English words well. Therefore, in future studies, using more complex topic modeling methods to unearth more hidden text structures should be considered.

**Author Contributions:** Conceptualization, J.S., A.M.M. and E.B.; methodology, J.S. and Q.Y.; software, Q.Y. and S.S.; validation, S.S. and J.S.; formal analysis, Q.Y.; investigation, Q.Y.; resources, Q.Y.; data curation, Q.Y.; writing—original draft preparation, J.S., Q.Y. and S.S.; writing—review and editing, A.M.M. and J.S.; visualization, Q.Y. and S.S.; supervision, A.M.M. and J.S.; project administration, J.S.; funding acquisition, J.S. and S.S.

**Funding:** This research was funded by ['the World-Class Discipline Construction and Characteristic Development Guidance Funds for Beijing Forestry University + Construction of Urban and Rural Human Settlements Ecological Environment in Beijing and Beijing-Tianjin-Hebei Region'], grant number [2019XKJS0316] and the APC was funded by [2019XKJS0316].

**Conflicts of Interest:** The authors declare no conflict of interest.

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
