# Peer review of "Museum Tourism 2.0: Experiences and Satisfaction with Shopping at the National Gallery in London"

_sustainability, doi:10.3390/su11247108_

Round 1
Reviewer 1 Report
I found the article very interesting, addressing a somewhat neglected issue of the tourism experience (the shopping experience at tourism attractions), and the application of the LDA technique to social media is interesting.
I would appreciate, for the benefit of readers, if authors could clearly separate the discussion section of the findings from the conclusion section, which is not there in the article. In this, a summary of the study could be presented along with the limitations of the research and suggestions for future research.
Very nice reading and a fine description of LDA.
Author Response
1.1 If authors could clearly separate the discussion section of the findings from the conclusion section, which is not there in the article.
Response 1: Thank you. We have rewritten the section 5 and added sub-titles.
1.2 In this, a summary of the study could be presented along with the limitations of the research and suggestions for future research.
Response2: Please see section 5.4.
Reviewer 2 Report
Good paper. Just a minor revision required.
Author Response
Thank you very much for your positive comments.
Reviewer 3 Report
This manuscript analyzes a topic very interesting but it should be completely rewritten because it is very confusing and it does not have flow.
Here somme comments to improve the paper:
-The autor/s should justify with cites many parts of the manuscript. For example, in the introduction and literature review there are many sentences without cites, for example, Page 1, lines 28-30: “Although there are many studies on shopping experiences….” “Content analysis has increasingly been applied in tourism research and recently….”
-The gap is highlited in introduction and literature review, it is better not to repeat the same idea.
-The introduction is too short and they should indicate why they study these objetives with the framework proposed by Kotler (2004). The authors should explicate how to achieve the objectives.
-The literature is not worth it to the manuscript aims. They should address the literature review according the objectives of this paper. For example, they use the Kotler (2004) framework and VADER sentiment analysis and the authors have not explained that in the literature review.
-The methodology has to be changed to explicate how they collect the data and the measurements that they utilize.
-The results are not relationed with the objectives.
-Discussion section has to be rewritten according the objectives of this manuscript.
I recommend to organize this paper according this manuscript:
Lee, M.; Lee, S.; Koh., Y. (2019). Multisensory experience for enhancing hotel guest experience: Empirical evidence form big data analytics. International Journal of Contemporary Hospitality Management. DOI 10.1108/IJCHM-03-2018-0263
Author Response
1. I recommend to organize this paper according this manuscript:
Lee, M.; Lee, S.; Koh., Y. (2019). Multisensory experience for enhancing hotel guest experience: Empirical evidence form big data analytics. International Journal of Contemporary Hospitality Management. DOI 10.1108/IJCHM-03-2018-0263
Response 1: Thank you for your recommendation. We have re-organized this version according to Lee, Lee, & Koh (2019).
2. The author/s should justify with cites many parts of the manuscript. For example, in the introduction and literature review there are many sentences without cites, for example, Page 1, lines 28-30: “Although there are many studies on shopping experiences….” “Content analysis has increasingly been applied in tourism research and recently….”
Response 2: We have added the cites, and move the sentence of line 30-31 to the literature review part.
3. The gap is highlighted in introduction and literature review, it is better not to repeat the same idea.
Response 3: We have deleted it in introduction.
4. The introduction is too short and they should indicate why they study these objectives with the framework proposed by Kotler (2004). The authors should explicate how to achieve the objectives.
Response 4: We have expanded the introduction in line 28-31, line 40-54, and line 57-64. The framework proposed by Kotler (2004) has been described in line 40-54 in this version.
5. The literature is not worth it to the manuscript aims. They should address the literature review according the objectives of this paper. For example, they use the Kotler (2004) framework and VADER sentiment analysis and the authors have not explained that in the literature review.
Response 5: The framework proposed by Kotler (2004) has been described in line 83-88 in this version. In this version, VADER sentiment analysis has been described in line 173-181. In the last version, the details have been explained in line 312-315 in section 4.3.
6. The methodology has to be changed to explicate how they collect the data and the measurements that they utilize.
Response 6: We have rewritten the section 3 and Figure 1 for the whole data process. Please see line 183-190. And, please see line 222-236 for details of the data collection, and line 238-243 for the details of data analysis.
7. The results are not related with the objectives.
Response 7: We have added a summary of results. Please see line 351-362 for the results related to the objectives.
8. Discussion section has to be rewritten according the objectives of this manuscript.
Response 8: We have rewritten it, and added sub-titles for section 5.2 to 5.4.
Round 2
Reviewer 3 Report
Thank you for the review